# Automated analysis of bacterial flow cytometry data with FlowGateNIST

**David Ross**●*

National Institute of Standards and Technology, Gaithersburg, Maryland, United States of America

* david.ross@nist.gov

## Abstract

Flow cytometry is commonly used to evaluate the performance of engineered bacteria. With increasing use of high-throughput experimental methods, there is a need for automated analysis methods for flow cytometry data. Here, we describe FlowGateNIST, a Python package for automated analysis of bacterial flow cytometry data. The main components of FlowGateNIST perform automatic gating to differentiate between cells and background events and then between singlet and multiplet events. FlowGateNIST also includes a method for automatic calibration of fluorescence signals using fluorescence calibration beads. FlowGateNIST is open source and freely available with tutorials and example data to facilitate adoption by users with minimal programming experience.

**Data Availability Statement:** All relevant data are within the manuscript and its Supporting Information files.

**Funding:** The author(s) received no specific funding for this work.

## Introduction

Flow cytometry has become one of the most commonly used methods for measuring the performance of engineered biological systems. It can provide rapid, high quality data for the distribution of engineered cellular response. A key step in flow cytometry data analysis is gating, or the discrimination between different cell sub-populations or between cells and non-cellular debris or other background events detected by the cytometer. Although gating is often performed manually, several approaches have been described for automated flow cytometry gating, which can improve reproducibility and decrease the time required for data analysis, particularly for high-throughput flow cytometry measurements [1–6].

In a typical bacterial flow cytometry measurement, the two most important gating steps are: 1) differentiation between cells and non-cell background events, and 2) differentiation between singlet cell events and doublet or higher-order mutliplet cell events. With mammalian cells, which have been the focus of most automated flow cytometry analysis methods, these two gating steps are simple. In particular, the relatively large size of mammalian cells makes them easy to distinguish from background events using the signals from the forward scatter and side scatter detectors of the cytometer. Consequently, most automated gating approaches have focused on the identification of different cell sub-populations based on multi-dimensional fluorescence signals obtained with flow cytometry [1, 2, 5, 6]. With smaller cells like bacteria, however, it can be more difficult to distinguish cells from background events and singlet events from doublet events. So, there is also a need for automatic gating methods that are

**Competing interests:** The authors have declared that no competing interests exist.

specifically aimed for use with those smaller cell types, and that provide a robust means to distinguish cells from background events and singlets from multiplets.

The FlowCal software package was first described using example data for engineered bacteria (*E. coli*) and yeast (*S. cerevisiae*) [3]. It includes an automatic gating algorithm that uses a 2D histogram binning method to find regions with the highest density of events in a two-dimensional plot (typically the side-scatter vs. forward-scatter signals). Regions with the highest density, encompassing a user-specified fraction of the total events, are gated as cell events, and regions with lower density are rejected as non-cell background events. The TASBE Flow Analytics software package is primarily focused on calibrating multiple fluorescence signals to comparable units, but it also includes automated gating, based on fitting with a Gaussian mixture model (GMM), to distinguish cells from background events [1]. Razo-Mejia *et al.* analyzed bacterial flow cytometry data with an automated gating method that uses a robust 2D Gaussian estimator to fit to the highest density region within a side-scatter vs. forward-scatter plot. With this approach, events within the central, high-density region (typically 40% of the total) are gated as cells, and events outside the central region are discarded as either background or multiplet events [7]. Many automated gating methods previously described for use with bacteria, including the three listed above, combine the cell vs. background and singlet vs. multiplet gating steps into a single step, using only the side-scatter and forward-scatter data.

Here, we present FlowGateNIST, a Python package for automated flow cytometry data analysis designed for use with small cells like bacteria and yeast. Automated flow cytometry data analysis with FlowGateNIST has four steps, described in more detail below: 1) Flow Cytometry Standard (FCS) data import. FlowGateNIST starts by converting data from the FCS format to a Pandas DataFrame for easy manipulation in Python. 2) Automated cell gating. FlowGateNIST uses a GMM approach and a comparison between measured cell samples and buffer blank samples for automated gating to discriminate between events that are most likely to be cells vs. events that are most likely to be background. 3) Automated singlet gating. FlowGateNIST then uses comparisons between the height, area, and width parameters of flow cytometry events to automatically discriminate between singlet and multiplet events. 4) Calibration of signals with fluorescent beads. In addition to automated gating, FlowGateNIST uses a multi-dimensional GMM applied to data for fluorescence calibration beads to convert measured fluorescence signals to comparable units.

## Results

### FlowGateNIST setup and data directory structure

FlowGateNIST is written in Python 3 and uses several Python packages, including NumPy [8], Pandas [9], SciPy [10], Matplotlib [11], and scikit-learn [12]. FlowGateNIST can be installed and used on any computer with an operating system capable of running Python 3. In addition, FlowGateNIST can be run with Jupyter Notebooks [13] within most web browsers to provide a record of the analysis for each dataset. The open-source FlowGateNIST repository on GitHub includes installation instructions, tutorials, and examples that should enable potential users to quickly adopt FlowGateNIST even without extensive Python experience (https://github.com/djross22/flowgatenist). However, for a brief and efficient overview of scientific data analysis with Python and the Python packages used by FlowGateNIST, we highly recommend the Python Data Science Handbook [14] by Jake VanderPlas, which is available online at https://jakevdp.github.io/PythonDataScienceHandbook/.

FlowGateNIST analyzes flow cytometry data in batches, and all FCS files for a batch need to be in the same data directory. In addition, to speed up the analysis, FlowGateNIST uses past results to initialize some of the analysis steps (see below). These past results are automatically

saved in an analysis memory directory that needs to be in the same directory structure as the data files. For these reasons, we recommend organizing the flow cytometry data in a directory structure as shown in Fig 1, with each flow cytometry experiment in a different sub-directory. The analysis memory directory will be automatically created by FlowGateNIST as a sub-directory in the user-specified top-level directory. In addition, we recommend the use of a "Jupyter Notebooks" sub-directory for each experiment to contain a record of the analysis for each experiment. Example Jupyter Notebooks for a typical analysis are included with the GitHub repository.

### Flow Cytometry Standard (FCS) data import

FlowGateNIST was inspired by the FlowCal package. It uses largely the same code for importing FCS data into Python. However, much of the code in the FlowCal library was aimed to create a data structure for flow cytometry with functionality similar to a Pandas DataFrame (FlowCal predated the Pandas package). In FlowGateNIST, we've used the relevant code from FlowCal to import FCS data directly into a Pandas DataFrame, with an associated metadata object. Apart from the use of Pandas DataFrames, the primary difference between FlowGate-NIST and FlowCal is that FlowGateNIST requires the measurement of blank samples, which it uses for automated cell gating (see below).

Automated data analysis with FlowGateNIST starts with a call to the fcs_to_dataframe() function. This function converts a batch of FCS files to FCSDataFrame objects and re-saves each of them using the pickle serialization module within Python. An FCSDataFrame object is simply a Pandas DataFrame object and associated metadata (e.g. acquisition start and end times). The fcs_to_dataframe() function has one required argument, data_directory, which indicates the directory where the FCS files are located. The function also requires an FCS file containing data measured for a buffer blank sample (indicated with the string "blank" somewhere in the file name). By default, the fcs_to_dataframe() function will parse the metadata in the FCS files and automatically choose the blank data file for the last blank sample measured before the cell samples. The user can alternately specify the blank file with the optional blank_-file argument. The function converts the blank FCS file as well as any FCS files for cell samples or calibration bead samples (indicated with the string "bead" somewhere in the file name). By default, the FCS files for cell samples are automatically detected as the list of all files with the ". fcs" extension and neither of the strings "blank" or "bead" in the file name. Additional optional arguments can be used to analyze or exclude FCS files for conversion, as described in the detailed documentation in the GitHub repository.

### Automated cell gating

FlowGateNIST uses a GMM approach, along with a comparison between a measured buffer blank sample and cell samples, for automatic gating. The buffer blank sample should be prepared the same way as the cell samples but without addition of cells, and it should ideally be measured just before the cell samples. As an example, for measurements of fluorescence protein expression, bacteria are typically diluted into phosphate-buffered saline (PBS) containing an antibiotic that halts translation (e.g. kanamycin or chloramphenicol). In that case, an appropriate buffer blank would be an aliquot of the same PBS with antibiotic. The buffer used for the blank and for cell dilutions should be freshly filtered to minimize particulates that are difficult to distinguish from the bacterial cells. To obtain a good estimate of the distribution of background events, we recommend the use or a larger acquisition volume for the buffer blank than for cell samples. For example, the background data shown in Fig 2A was obtained with a

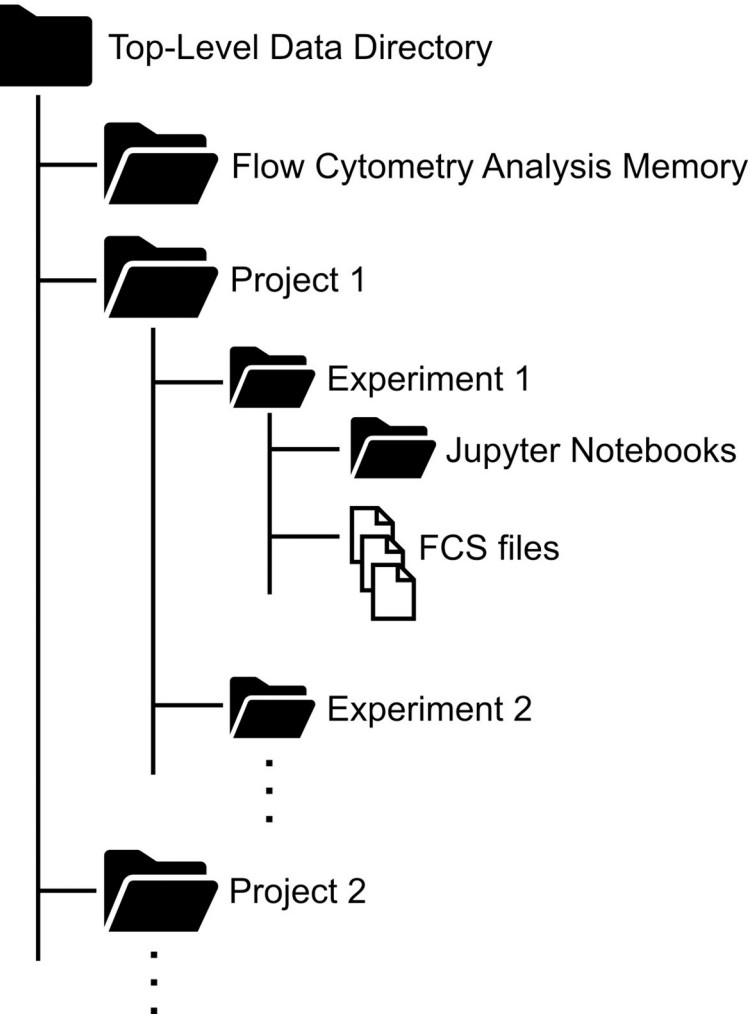

**Fig 1. Recommended data directory structure for FlowGateNIST.** FlowGateNIST analyzes data in batches. All FCS files for a batch must be contained in the same sub-directory. So, we recommend the use of separate sub-directories for each project and experiment as shown. The Flow Cytometry Analysis Memory directory will be automatically created by FlowGateNIST as described in the text and in the tutorial included as part of the GitHub repository.

150 μL acquisition volume while the cell data shown in Fig 2B was obtained with a 25 μL acquisition volume.

Automated cell vs. background gating is run via a call to the background_subtract_gating() function. The automated gating algorithm starts by fitting a GMM to the forward scatter and side scatter data for a buffer blank sample. It uses the base-ten logarithm of the height parameter for events detected in the forward and side scatter channels of the cytometer (i.e. the "FSC-H" and "SSC-H" channels). The number of mixture model components (i.e. clusters) used for fitting the background data is an optional argument of the background_subtract_gating() function. By default, the function starts by determining the optimal choice for the number of background clusters using the Bayesian information criterion (BIC) [14]. Optimization based on the BIC requires several repetitions of the GMM fit, which increases analysis time. So, we only recommend it for data collected with a new cytometer or new cytometer settings. Once the optimal number of background clusters is known, it can be specified with the num_back_clusters argument to reduce the analysis time. The distribution of background events in

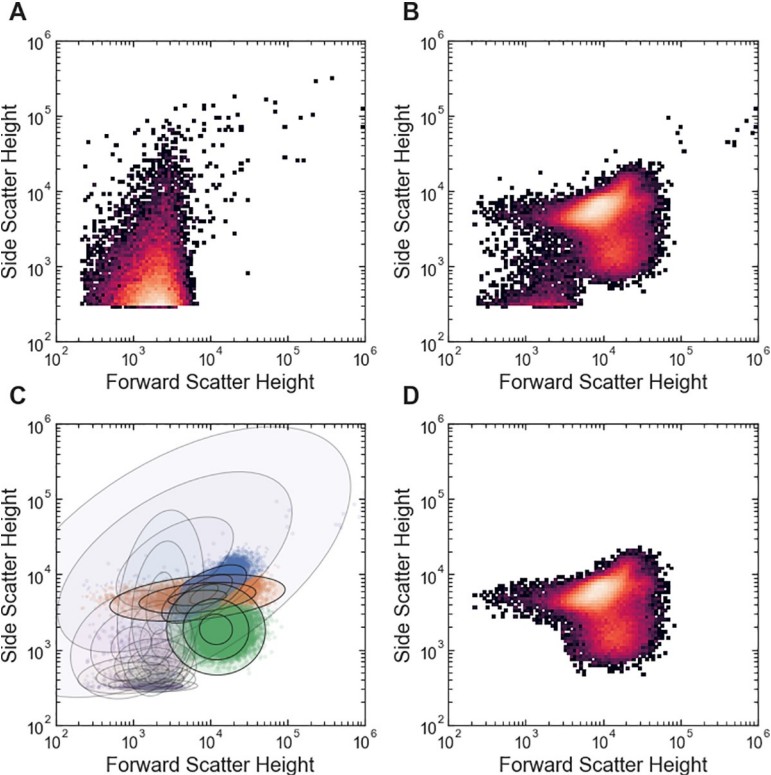

**Fig 2. Example of automated cell vs. background gating with FlowGateNIST.** (A) 2D histogram of side scatter vs. forward scatter for buffer blank sample. (B) 2D histogram of side scatter vs. forward scatter for *E. coli* cell sample. (C) Scatter plot showing Gaussian mixture determined during automated gating. The blue, orange, and green clusters are the cell events. The purple cluster is the background events. The ellipses show the contours for 1, 2, and 3 standard deviations for each Gaussian mixture component. (D) 2D histogram of side scatter vs. forward scatter for the cell sample in (B) after automatic gating to remove background events.

the side scatter vs. forward scatter plot is typically asymmetric and bordering on the lower edge of the plot, at the threshold value set for cytometry event collection. So, the optimal number of background clusters can be fairly high. For example, in the analysis of *E. coli* data, we typically use eight background clusters (Fig 2A).

After fitting the blank sample data to determine the background distribution, the background_subtract_gating() function performs a second GMM fit using data from the cell samples. As with the background fit, this GMM fit uses the base-ten logarithm of the height parameter for the forward and side scatter signals. The mixture model for this second GMM fit includes the clusters resulting from the background fit plus additional clusters for the cell events. In the GMM fit algorithm, the means, covariances, and relative weights of the background clusters remain fixed at the values determined by the fit to the blank sample data, and only the parameters of the cell clusters are allowed to vary to fit the data. Thus, the position and shape of the background distribution is maintained during the fit while the parameters for the cell clusters and the relative weights between the background distribution and cell clusters are varied to find the optimal GMM fit. The algorithms used for GMM fits are not guaranteed to find the globally optimal fit results [14]. So, in practice, GMM fits are typically run multiple times with different random initializations, and the best overall result is kept. The number of random initializations used for the GMM fit is another optional argument of the background_subtract_gating() function. With a higher number of random initializations, the

algorithm takes longer, but the probability of a poor fit result is lower. GMM fits can also be initialized non-randomly. In particular, the result from a prior fit to similar data can often be used to find the best overall fit more rapidly than multiple random initializations. FlowGate-NIST uses a combination initialization approach: First, the GMM fit algorithm is run with a user-specified number of random initializations. In addition, all successful GMM fit results are saved in an analysis memory directory, and a user-specified number of results from the analysis memory are also used to initialize each new GMM fit. For each call to the background_subtract_gating() function, the best fit result across all random and non-random initializations is used. Different sets of analysis memory results can be saved for different cell types or instrument settings, using the cell_type argument. The number of analysis memory results that can be saved is limited only by the capacity of the system where the flow cytometry data is stored.

The background_subtract_gating() function analyzes data in batches and applies the same gate to all data files within a batch. By default, the function treats all cell data files in the data directory as a single batch. The optional argument, samples, can be used to specify a sub-set of the cell data files to be analyzed as a batch (see the tutorial in the GitHub repository). The background subtraction GMM fit uses a concatenation of all of the cell data in the batch. The background_subtract_gating() function automatically produces a set of diagnostics plots that can be used for adjusting the function arguments. These plots include 2D histograms of the data before and after gating for every sample in each batch (e.g. Fig 2B and 2C) and color-coded scatter plots showing the GMM cluster assignments.

## Automated singlet gating

The next step in automated cytometry data analysis with FlowGateNIST is automated gating to distinguish between singlet events and doublet or higher-order multiplet events. Multiplet events occur when multiple cells cross the flow cytometer detection volume simultaneously. Because the aim of cytometry analysis is to measure the properties of single cells, multiplet events need to be distinguished and ignored for an accurate result. FlowGateNIST includes two functions for singlet vs. multiplet gating: the singlet_gating() function and the singlet_gating_width() function. Both singlet gating functions use GMM fits to analyze batches of cytometry data files, with arguments similar to those used for the background_subtract_gating() function. As with the automated cell gating, the singlet gating functions use a combination of random initializations and initializations from an analysis memory.

The singlet_gating() function uses a 2D GMM fit to the base-ten logarithm of the side-scatter height parameter vs. the base-ten logarithm of the ratio of the side-scatter height to side-scatter area parameters. With the Attune NxT flow cytometer used for the examples described here, the instrument is automatically calibrated so that the height and area parameters are the same for singlet events. The singlet_gating() function takes advantage of this calibration to select singlets as events that belong to clusters with a mean log-ratio of the height to area close to zero (Fig 3). The optional singlet_mean_center argument can be used to specify a different central value for singlet clusters to facilitate use with other cytometry instruments.

The singlet_gating_width() function uses a 2D GMM fit to the base-ten logarithm of the side-scatter height parameter vs. the base-ten logarithm of the side-scatter width parameter. The function selects singlets as events that belong to clusters centered at a log-height below a cutoff value. By default, the cutoff is automatically set at 1.4 times the center value for the cluster with the most detected events. The cutoff can also be set manually using the optional singlet_width_cutoff argument.

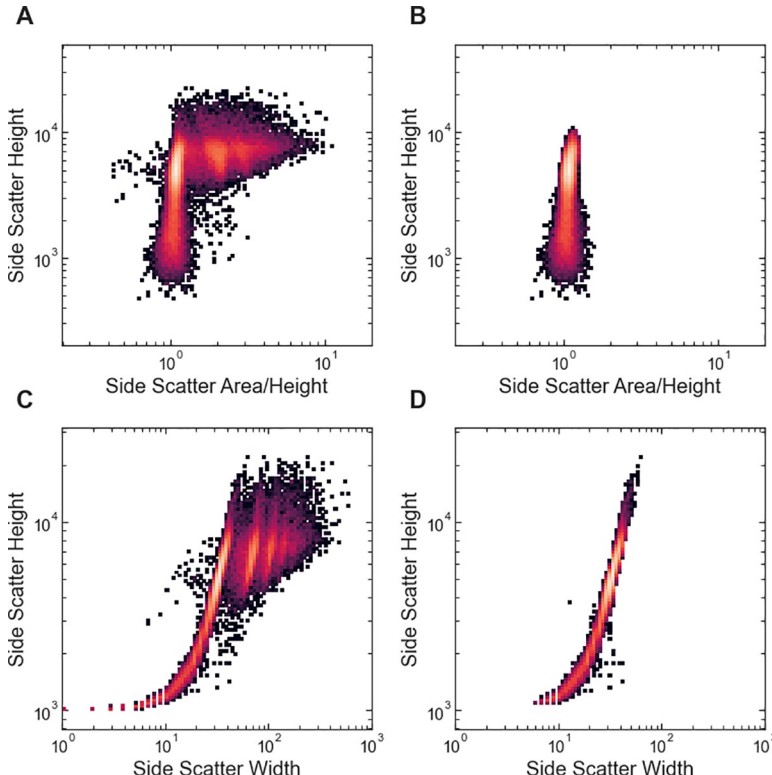

**Fig 3. Examples of automated singlet gating with FlowGateNIST.** (A) 2D histogram of the side scatter height vs. the ratio of side scatter area to height. The flow cytometry instrument used to collect the data is automatically calibrated so that the area to height ratio is approximately one for singlet events. (B) 2D histogram of the data shown in (A) after automatic singlet gating using the singlet_gating() function. (C) 2D histogram of the side scatter height vs. the side scatter width. (D) 2D histogram of the data shown in (C) after automatic singlet gating using the singlet_gating_width () function.

Both singlet gating functions automatically generate a set of diagnostics plots including 2D histograms of the data before and after singlet gating for every sample in each batch (e.g. Fig 3) and color-coded scatter plots showing the GMM cluster assignments.

The two-step cell and singlet gating approach used by FlowGateNIST takes advantage of the full set of event parameters provided by most modern flow cytometers (height, area, and sometimes width). As a result, it can give a better representation of the distribution of single cell phenotypes. For example, a single-step gating approach that uses only the height parameter from the forward and side scatter signals for combined cell and singlet gating undercounts the number of singlet cell events and misses a distinct cell sub-population (Fig 4).

## Calibration of fluorescence signals with beads

As with the FlowCal and TASBE Flow Analytics software packages, FlowGateNIST includes functions for calibration of fluorescence data using fluorescent beads. A variety of fluorescence calibration beads are now available, and some of them are NIST traceable [15, 16]. For bacterial work, we use Sphereotech Rainbow Calibration Particles, catalog number RCP-30-5A, because they have fluorescence intensities comparable to that found in bacteria engineered to express fluorescent proteins. That calibration bead set is a mixture of eight different beads with varying fluorescence intensities that are calibrated to the molecular equivalents for five

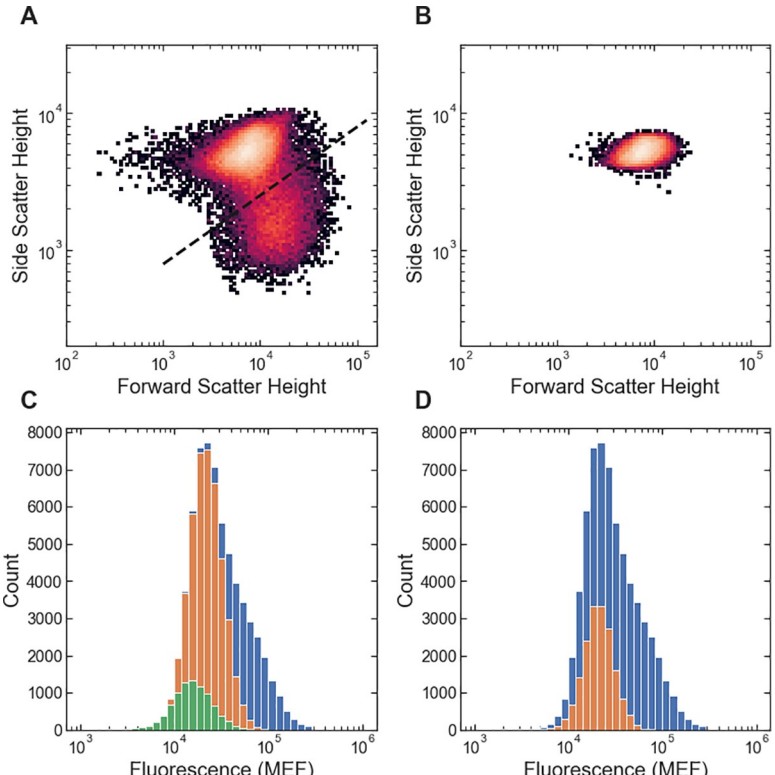

**Fig 4. Comparison between the two-step cell and singlet gating of FlowGateNIST and a single-step gating approach.** (A) Results of the FlowGateNIST two-step gating are shown in a 2D histogram of the side scatter vs. forward scatter signals. Only singlet cell events are plotted (as determined using the singlet_gating() function). (B) Results of a single-step gating approach that uses only the height parameter from the forward and side scatter signals to select the region of the plot with the highest density. (C) Florescence signal histograms corresponding to (A). The blue histogram bars show the distribution of fluorescence signal for all cell events. The orange histogram bars show the fluorescence for the singlet events plotted in (A). The green histogram bars show the fluorescence for the singlet cells in the sub-population below the bold dashed line in (A). That sub-population has a distinctly different scattering phenotype and a slightly, but significantly lower fluorescence than the main cell population. (D) Florescence signal histograms corresponding to (B). The blue histogram bars show the distribution of fluorescence signal for all cell events. The orange histogram bars show the fluorescence for the events plotted in (B). The single-step gating approach undercounts the number of singlet cells and completely misses the sub-population below the bold dashed line in (A). Increasing the size of the gate used for the single-step approach increases the number of singlet cells correctly gated but also includes an increased number of doublet evets.

different fluorescent dyes. To collect bead data, we typically analyze a sample of 200 μL focusing fluid with one drop of calibration bead solution.

Bead calibration in FlowGateNIST is run via a call to the fit_bead_data() function. That function starts with an automated algorithm to gate the bead data for singlet bead events. After the singlet bead events are identified, the function uses a 2D GMM fit to the fluorescein (FITC) and phycoerythrin (PE) channels of the flow cytometer ("BL1" and "YL1" on the Attune NxT cytometer) to identify clusters of events corresponding to the different calibration beads in the set (Fig 5). The multi-dimensional GMM fit used by FlowGateNIST for bead cluster identification can distinguish the full set of beads more effectively than methods that use only a single fluorescence channel. Consider, for example, the data shown in Fig 5: The first three bead clusters are partially overlapping. Using only a single fluorescence channel (Fig 5D), the first two clusters are completely unresolvable, but they can be distinguished with the 2D GMM fit used by FlowGateNIST (Fig 5A). After identifying the different bead clusters, the

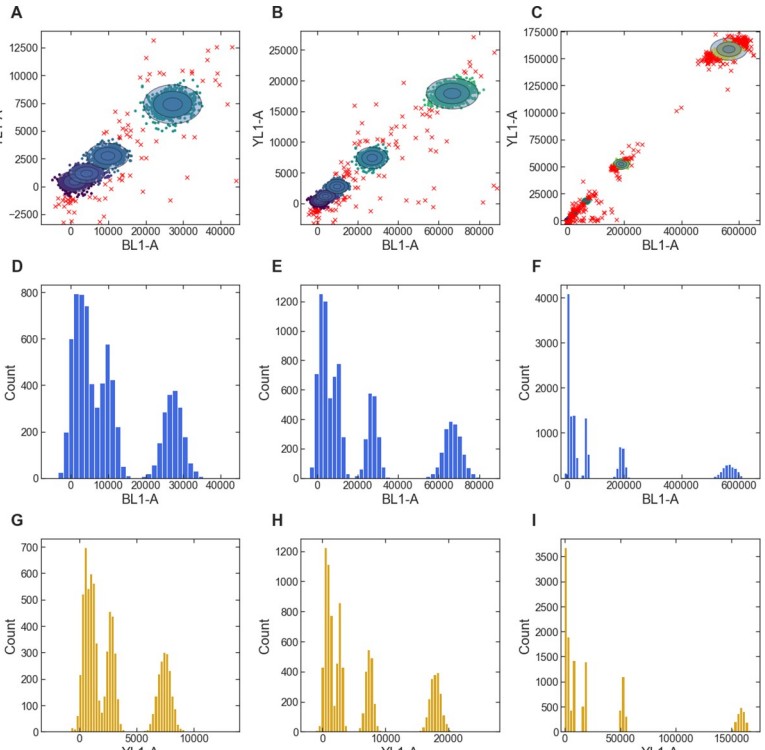

**Fig 5. Example of bead-based fluorescence calibration with FlowGateNIST.** (A-C) 2D GMM fit of the YL1-A vs. BL1-A fluorescence signals from calibration beads. Only data for singlet bead events is shown. Bead clusters are shown as different colored points and shaded ellipses. Outlier points ignored by the calibration are plotted as red X's. The three plots show the same data with different scales. (D-F) Histograms of the BL1-A signal from calibration beads. Data are the same as in (A-C), with outliers excluded. The two bead clusters with the lowest signal are completely unresolvable with the BL1-A signal. (G-I) Histograms of the YL1-A signal from calibration beads. Data are the same as in (A-C), with outliers excluded.

fit_bead_data() function excludes outlier events and determines the mean fluorescence values for each bead cluster. The function then uses a linear least-squares fit of the mean values vs. the vendor-supplied calibration values to determine the calibration.

The function fit_bead_data() generates several diagnostic plots, including 2D scatter plots of the measured fluorescence values overlaid with the GMM fit results and plots of the calibration functions for each fluorescence channel. More details for adjusting the function arguments are described in the tutorial included in the GitHub repository.

## Discussion

FlowGateNIST is free and open source, and it is programmed in Python, one of the most widely used programming languages for data analysis. The code is maintained in a publicly available repository on GitHub. In addition to the functions described here that are meant to be applied as the first data analysis steps for all bacterial flow cytometry experiments, the Flow-GateNIST package includes additional functions and example code to facilitate a complete analysis of, for example, high-throughput measurements of bacterial dose-response curves. Tutorials illustrating a complete flow cytometry analysis are included with the GitHub repository. Finally, the diagnostic plots generated by FlowGateNIST are automatically saved in pdf files and also displayed in an interactive Python environment such as Jupyter Notebooks, providing both a record of the analysis that was performed and a route to optimize the analysis.

## Materials and methods

All example data was obtained using *E. coli* strain MG1655Δ*lac* [17] containing a plasmid in which a variant of the *E. coli lac* repressor, LacI, controls the expression of Enhanced Yellow Fluorescent Protein (YFP) in response to induction by isopropyl-β-D-thiogalactoside (IPTG). The plasmid sequence is available at https://www.ncbi.nlm.nih.gov/nuccore/ MT702634. The LacI variant used for the example data has an inverted dose-response curve, i.e. YFP expression is high at zero IPTG, and decreases as the IPTG concentration is increased [18].

An automated growth protocol was used for all example data [18]. Briefly: *E. coli* cultures were grown in a rich M9 media (3 g/L KH2PO4, 6.78 g/L Na2HPO4, 0.5 g/L NaCl, 1 g/L NH4Cl, 0.1 mmol/L CaCl2, 2 mmol/L MgSO4, 4% glycerol, and 20 g/L casamino acids) supplemented with 50 μg/mL kanamycin. Cultures were grown in clear-bottom 96-well plates with 1.1 mL square wells (4titude, cat. #4ti-0255), with 500 μL culture volume per well. A multi-mode plate reader (BioTek, Neo2SM) was used to incubate cultures at 37°C with a 1°C gradient applied from the bottom to the top of the incubation chamber to minimize condensation on the inside of the membrane. During incubation, the plate reader was set for double-orbital shaking at 807 cycles per minute. Cultures were first grown to stationary phase (12 hours) in media without IPTG, then diluted 50-fold into media containing different concentrations of IPTG and grown for 160 minutes. Cultures were then diluted 10-fold into media with IPTG and grown again for 160 minutes. An automated plate sealer (4titude, a4S) was used to seal each 96-well plate with a gas permeable membrane (4titude, cat. #4ti-0598) to minimize evaporation during growth. The gas permeable membrane was automatically removed from each 96-well plate after incubation using an automated de-sealer (Brooks, XPeel). For the example data shown in Figs 2–4, zero IPTG was used for all three growth periods.

After the 2nd 160 minute growth, 5 μL of cell culture was diluted into 195 μL of PBS with 170 μg/mL chloramphenicol. The samples in PBS were then incubated at room temperature for between 30 minutes and 60 minutes before flow cytometry measurements. Flow cytometry measurements were performed using an Attune NxT flow cytometry with autosampler. A buffer blank sample (200 μL of PBS with 170 μg/mL chloramphenicol) was measured just before the cell samples. A calibration bead sample was measured just after the cell samples. The bead sample was prepared by diluting one drop of Sphereotech Rainbow Calibration Particles, catalog number RCP-30-5A, into 200 μL of focusing fluid. The forward scatter and side scatter thresholds were set so that all cell events were detected while the number of background events was minimized. The acquisition volume was set to 150 μL for the buffer blanks and bead samples and 25 μL for the cell samples. With this acquisition volume, approximately 20,000 singlet cells were detected in each cell sample. The sample flow rate was set to 100 μL/min for all samples. The autosampler was set to mix each sample three times by repeated aspiration and to run three rinse cycles between samples.

## Supporting information

**S1 File. The file "Supplementary Data.zip" contains the raw data used for Figs 2–5 as well as the processed data required to reproduce the plots.**
(ZIP)

## Acknowledgments

We would like to thank Vanya Paralanov, Brian DeCoste, and Peter Tonner for thoughtful discussions and help with Python code during the development of FlowGateNIST. We would also

like to thank Drew Tack for the example data used in this manuscript, and Sebastian Castillo-Hair and John Sexton for suggesting the name "FlowGateNIST".

## Author Contributions

**Conceptualization:** David Ross.

**Formal analysis:** David Ross.

**Investigation:** David Ross.

**Methodology:** David Ross.

**Software:** David Ross.

**Writing – original draft:** David Ross.

**Writing – review & editing:** David Ross.

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
