## [Decision Letter · Decision Letter 0]

26 May 2021

PONE-D-21-11845

Automated Analysis of Bacterial Flow Cytometry Data with FlowGateNIST

PLOS ONE

Dear Dr. Ross,

Thank you for submitting your manuscript to PLOS ONE. After careful consideration, we feel that it has merit but does not fully meet PLOS ONE’s publication criteria as it currently stands. Therefore, we invite you to submit a revised version of the manuscript that addresses the points raised during the review process.

As you can see, both reviewers  agree that the new analysis tools will be useful for the community.  Please adress the minor concerns of reviewer 2. 

We look forward to receiving your revised manuscript.

Kind regards,

Patrick Lajoie, PhD

Academic Editor

PLOS ONE

Journal Requirements:

Reviewers' comments:

Reviewer's Responses to Questions

**Comments to the Author**

1. Is the manuscript technically sound, and do the data support the conclusions?

Reviewer #1: Yes

Reviewer #2: Yes

2. Has the statistical analysis been performed appropriately and rigorously? 

Reviewer #1: N/A

Reviewer #2: N/A

3. Have the authors made all data underlying the findings in their manuscript fully available?

Reviewer #1: Yes

Reviewer #2: Yes

4. Is the manuscript presented in an intelligible fashion and written in standard English?

Reviewer #1: Yes

Reviewer #2: Yes

5. Review Comments to the Author

Reviewer #1: The authors present work showing open source software for gating bacteria and provide examples of how the software works as well as examples of the program working. The figures presented show automated gating, background subtraction as well as singlet analysis. The usefulness of using beads for automatic calibration is a useful feature that has been previously used in flow cytometry of extra cellular vesicules.

The authors present work that is clear and of value to people to use, the links to guidebooks and resources will be of value to people reading this article.

Reviewer #2: In this study, Ross D presents a Python package for automated flow cytometry data (FlowGateNIST), designed to analyze small cells like bacteria and yeast. This package has several advantages: to differentiate between cells and background events, singlet and multiple events, automatic calibration of fluorescence signals (trough fluorescence calibration beads), and not least important; it is freely available. In this reviewer's opinion, the publication of this work is relevant inside the flow cytometry field because it is a technique used worldwide with both clinical and research application. I have minors comments, and I will appreciate if the author can add the information in a new version of the manuscript.

Minor comments

1. Please clarify if the analysis memory is limited.

2. Please clarify if FlowGatesNIST can be open in any operative system.

3. Please clarify if FlowGatesNIST can be open in any internet browser.

4. The FlowCal package inspires FlowGatesNIST, and you provided essential technical information in the result section. However, please include a paragraph to be more explicit (and easier) about the differences between this new package and FlowCal. This new information should be writing thinking in the final user, who usually is biomedical researchers or medical technicians.

6. PLOS authors have the option to publish the peer review history of their article (what does this mean?). If published, this will include your full peer review and any attached files.

Reviewer #1: No

Reviewer #2: **Yes: **Leslie Chavez-Galan

---

## [Author Response · Author response to Decision Letter 0]

15 Jun 2021

Point-by-Point Response:

Reviewer #2: 

1. Please clarify if the analysis memory is limited.

This clarification was added (lines 177-179 in the revised manuscript):

“The number of analysis memory results that can be saved is limited only by the capacity of the system where the flow cytometry data is stored.”

2. Please clarify if FlowGatesNIST can be open in any operative system.

This clarification was added (lines 71-72 in the revised manuscript):

“FlowGateNIST can be installed and used on any computer with an operating system capable of running Python 3.”

3. Please clarify if FlowGatesNIST can be open in any internet browser.

Python packages are not normally run directly in an internet browser. But Jupyter Notebooks can be used to run FlowGateNIST (and other Python packages) within a web browser. An appropriate clarification was added (lines 72-74 in the revised manuscript):

“In addition, FlowGateNIST can be run with Jupyter Notebooks [13] within most web browsers to provide a record of the analysis for each dataset.”

4. The FlowCal package inspires FlowGatesNIST, and you provided essential technical information in the result section. However, please include a paragraph to be more explicit (and easier) about the differences between this new package and FlowCal. This new information should be writing thinking in the final user, who usually is biomedical researchers or medical technicians.

That additional information on the difference between FlowGateNIST and FlowCal (lines 101-103 in the revised manuscript):

“Apart from the use of Pandas DataFrames, the primary difference between FlowGateNIST and FlowCal is that FlowGateNIST requires the measurement of blank samples, which it uses for automated cell gating (see below).”

---

## [Editor Report · Decision Letter 1]

22 Jun 2021

Automated analysis of bacterial flow cytometry data with FlowGateNIST

PONE-D-21-11845R1

Dear Dr. Ross,

Thank you for submitting your revised manuscript. I was able to assess the revised version based on the previous reviewers  comments. We’re pleased to inform you that your manuscript has been judged scientifically suitable for publication and will be formally accepted for publication once it meets all outstanding technical requirements.

Kind regards,

Patrick Lajoie, PhD

Academic Editor

PLOS ONE

---

## [Editor Report · Acceptance letter]

26 Jul 2021

PONE-D-21-11845R1 

Automated analysis of bacterial flow cytometry data with FlowGateNIST 

Dear Dr. Ross:

I'm pleased to inform you that your manuscript has been deemed suitable for publication in PLOS ONE. Congratulations! Your manuscript is now with our production department. 

Kind regards, 

on behalf of

Dr. Patrick Lajoie 

Academic Editor

PLOS ONE